# Prenatal Diagnosis of Vaginal Ectopic Ureter Insertion—Case Outcome and Literature Overview

**DOI:** 10.3390/diagnostics15141788

**Published:** 2025-07-16

**Authors:** Iulian Gabriel Goidescu, Georgiana Nemeti, Adelina Staicu, Mihai Surcel, Cerasela Mihaela Goidescu, Ioana Cristina Rotar, Gheorghe Cruciat, Daniel Muresan

**Affiliations:** 1Obstetrics and Gynecology I, Mother and Child Department, University of Medicine and Pharmacy “Iuliu Hațieganu”, 400006 Cluj-Napoca, Romania; goidescu.iulian@elearn.umfcluj.ro (I.G.G.); mihai.surcel@elearn.umfcluj.ro (M.S.); cristina.rotar@umfcluj.ro (I.C.R.); gheorghe.cruciat@elearn.umfcluj.ro (G.C.); muresandaniel01@elearn.umfcluj.ro (D.M.); 2Department of Internal Medicine, Medical Clinic I—Internal Medicine, Cardiology and Gastroenterology, University of Medicine and Pharmacy “Iuliu Hațieganu”, 400006 Cluj-Napoca, Romania; sava.cerasela@elearn.umfcluj.ro

**Keywords:** single site ectopic ureter, prenatal diagnosis, cystic renal dysplasia, OHVIRA

## Abstract

**Background and clinical significance:** Ectopic ureters are a rare urinary tract malformation, typically diagnosed in childhood and infrequently in adulthood. The prenatal detection by ultrasound and magnetic resonance imaging (MRI) of this clinical entity has scarcely been reported. Careful foetal scanning during the late second and third trimester might provide clues and lead to prenatal detection. However, even the postnatal diagnosis is challenging, and often delayed towards adulthood, since the condition may present with nonspecific symptoms, leading to underdiagnosis or misdiagnosis. In female patients, approximately 25% of ectopic ureters open into the vagina. Due to the high risk of recurrent urinary tract infections and the potential development of uretero-hydronephrosis, timely diagnosis is essential, and prompt surgical correction is mandated. **Case presentation:** We report the case of a 33-year-old GII PI patient diagnosed with cystic dysplasia of the left foetal kidney at the 16 WG (weeks of gestation) scan. The malformation was consistent at 21 WG when karyotyping by amniocentesis identified a normal female molecular karyotype. MRI performed at 28 weeks confirmed the left renal dysplasia and raised the suspicion of an abnormal insertion of the left ureter into the vagina. After delivery, the vaginal ureteral ectopy was confirmed at 3 weeks postpartum via cystoscopy. Postpartum whole exome sequencing identified a variant of uncertain significance (VUS) mutation in the *SOX 13* gene (*SRY*-box transcription factor 13). Renal scintigraphy performed 7 months postnatally identified a hypo/afunctional left kidney which led to the indication of nephrectomy by the paediatric urologist. The surgical intervention was performed at 8 months postpartum with a favourable outcome. **Conclusions:** Ectopic ureters are a pathology generating life-long morbidity and discomfort of the offspring and young adult. Awareness to this pathology must be raised among clinicians, especially regarding the potential detection by minute prenatal ultrasound examinations, followed by MRI to refine diagnosis. Postnatally, the persistence of suspicious yet unspecific symptoms, in both males and females, must trigger thorough imaging/cystoscopic examination to reach diagnosis and provide correct management.

## 1. Introduction

Congenital anomalies of the kidney and urinary tract (CAKUT) represent approximately 20–30% of all congenital anomalies detected prenatally and the leading abnormalities identified during third trimester scans [1,2].

CAKUT represent a complex array of abnormalities with various degrees of clinical impact on the foetus, from those who require mere surveillance to those who deem medical treatment, surgery, to lethal malformations. Cases with rare and intricate anomalies require imaging and genetic studies to achieve an accurate diagnostic and to provide proper counselling to parents, usually in a multidisciplinary team approach.

The ectopic ureter, an infrequent CAKUT with an incidence of 0.025–0.05% in the general population, is a condition where the ureter fails to open in the trigone area of the bladder [3,4,5].

The ectopic opening may be found at the level of the cervix, uterus, fallopian tube, bladder neck and upper urethra, urethral septum, vaginal vestibule, Gartner cyst, and rectum in newborn females, and the posterior urethra, prostate tube, seminal vesicles, and ejaculatory duct in newborn males [6,7,8,9,10]. It can be found in 20% of cases as a single-system ectopic ureter (SSEU), but it is more commonly associated with the complete duplex kidney system, in 80–85% of cases [3,7]. It is more frequently diagnosed in female foetuses with a ratio of 2:1–6:1 = F–M [11].

There are regional variations in the prevalence of ectopic ureters in a single collecting system, and the underling cellular and molecular mechanisms leading to developmental ureter implantation abnormalities are not fully understood [6,12].

Patient presentation depends on the insertion site of the ectopic ureter and may include continuous urinary incontinence (with normal voiding pattern, more common in females due to the ectopic ureter bypassing the sphincter mechanism), recurrent urinary tract infections, abdominal or flank discomfort (if associated with hydronephrosis or infection), and poor renal function in the affected kidney.

Prenatal diagnosis is rare, while most cases are diagnosed in childhood, as the condition often presents with the above mentioned non-specific and difficult-to-interpret symptoms, particularly in children under 1 year of age [7]. SSEU has been established as a distinct clinical entity which must be suspected in girls with incontinence and dribbling or in boys with epididymitis/genitourinary complaints and a non-functioning kidney after imaging investigations [13].

Given both the quality of the latest ultrasound equipment and the improved scanning techniques of the past years, prenatal ultrasound may raise the suspicion of a renal anomaly or pyelocalyceal duplication, with the adjunct of foetal magnetic resonance imaging (MRI) to improve accuracy or refine diagnosis. Genetic testing performed during pregnancy or after delivery may rule out or confirm a genetic background [14,15].

CAKUT represent a wide array of pathologies, some infrequent and difficult to pinpoint. Prenatal diagnosis is crucial to facilitate case tailored family counselling and to plan for prompt and appropriate interventions.

This is the presentation of a case of an ectopic ureter with a vaginal opening which benefitted from an early third trimester prenatal diagnosis.

## 2. Case Presentation

This is the case of a 33-year-old G2P1A0 female patient, with an unremarkable general, gynaecologic, and obstetric history (one prior term delivery by C-section due to ophthalmologic indication). The patient attended all the recommended prenatal visits and performed the required investigations during the first trimester. First trimester genetic screening for Down syndrome by the integrated test was performed and provided a low-risk result.

At the 17 weeks of gestation (WG) scan, the suspicion of a dysplastic left kidney was raised without any other associated abnormality (Figure 1). This finding warranted further evaluation at 20 WG to reassess kidney development and function. Given the potential association of renal dysplasia with genetic or syndromic conditions, invasive prenatal diagnosis by amniocentesis was offered and declined by the couple at this time.

At 21 WG the suspicion of a dysplastic left kidney persisted, and the couple agreed to amniocentesis which revealed a normal female molecular karyotype (46XX) and a negative panel for cystic renal dysplasia—the Blueprint Genetics, Finland, Polycystic Kidney Disease Panel (version 4, 30 October 2021) Plus Analysis (*DNAJB11*, *DZIP1L*, *GANAB*, *HNF1B*, *JAG1*, *LRP5**, *NOTCH2**, *PKD1**, *PKD2*, *PKHD1*, *PRKCSH*, *SEC61A1*, *SEC63*). After interdisciplinary counselling and discussions with the family, the pregnancy was continued according to the planned follow up schedule.

The 28 WG ultrasound evaluation re-established the diagnosis of a dysplastic left kidney, and a foetal MRI (magnetic resonance imaging) was scheduled to refine the diagnosis. The MRI confirmed the left renal cystic dysplasia, most likely secondary to an ectopic ureter with a vaginal insertion; the right kidney exhibited a cortical cyst, without pyelocalyceal dilation and normal-appearing parenchyma (a duplication of the collecting system could not be ruled out with certainty) (Figure 2).

The couple were counselled by a multidisciplinary team regarding the prognosis of their offspring, and the decision to continue pregnancy was reached, with a planned postnatal follow-up including renal ultrasound, renal scintigraphy, and possibly voiding cystourethrography or MRI urography to assess the renal function, to confirm the anatomy of the urinary tract, and to determine the need for surgical intervention in case an obstructive ectopic ureter was diagnosed. Management, according to the findings, including paediatric, nephrological, and urologic surgical involvement as required, was explained to and accepted by the parents.

Subsequent dynamic ultrasound examinations during the third trimester found corresponding images of the left kidney with a similar polycystic appearance and a right renal cyst of constant dimensions (Figure 3).

The patient delivered electively at 39 WG by C-section a newborn female weighing 3550 g, with an Apgar score of 10.

At 3 weeks postpartum, an examination by the paediatric nephrologist confirmed the presence of an ectopic vaginal ureteral opening, as identified by the foetal MRI. To better characterize the infant’s condition, genetic testing using whole exome sequencing (WES) analysis was indicated, which identified only a variant of uncertain significance (VUS) mutation in the *SOX 13* gene (c.1009_1010insG, p.(Pro337Argfs*10) (*SRY*-box transcription factor 13) (Figure 4).

Exploratory urethrocystoscopy performed at 3 months postnatally described a hyperaemic bladder, with a normally positioned right ureteral opening present at the level of the bladder trigone, and a well-configured, but noted to be slightly gaping, absent left ureteral opening, with gradual loss of the inter-urethral band (Figure 5A,B).

The gaping right ureteral opening could be indicative of vesicoureteral reflux, which should be evaluated further to prevent kidney damage. For this reason, a voiding cystography was scheduled but no signs of vesicoureteral reflux were detected on this occasion.

Renal scintigraphy performed 7 months postnatally revealed a right kidney with a normal parenchyma (approx. 5.5 cm in the long axis), pyelocalyceal hypotonia, without signs of obstruction. The left kidney was hypo/afunctional (Figure 6).

Due to the absent function of the left kidney and taking in to account the postpartum evolution lined with recurrent urinary tract infections, the parents were counselled that prophylactic nephrectomy was indicated. Surgery was performed at 8 months postnatally, and the pathology examination confirmed the imaging suspicion of a dysplastic kidney.

The postsurgical evolution was non-remarkable, without any episodes of urinary tract infection and normal child development.

## 3. Discussion

CAKUT are a heterogeneous group of abnormalities of the kidneys and outflow tracts with a large spread in its estimated prevalence of 4–60 in 10,000 births [16]. Extra-renal anomalies are not uncommon and complicate the prognosis in one third of infants with CAKUT [1].

The prenatal ultrasound diagnosis of CAKUT can be achieved as early as the first trimester scan for some malformations, to a peak detection rate during the second trimester morphology scan, and a later case pick-up during the third trimester for late-onset anomalies. Foetal MRI is routinely used for complex cases to refine the diagnosis and to aid in the patient outcome estimation and parent counselling.

Prenatal suspicion of an ectopic ureter opening into the vagina is challenging due to its rarity, with ultrasound findings primarily reflecting secondary renal damage. Occasionally, an ectopic ureter is associated with additional abnormalities, such as renal dysplasia and various non-genitourinary anomalies, including congenital heart disease, spinal cord malformations, and anorectal malformations. These associations suggest that ectopic ureters may result from broader developmental disturbances during embryogenesis [17,18].

### 3.1. Anatomic Considerations

The course of the ectopic ureter generally corresponds to that of a normal ureter, differences occurring mostly at the lower insertion site, especially in the presence of simultaneous ectopic malformations (duplex ureter or retrocaval ureter) [11]. This is why SSEU and ectopy in the presence of a duplex draining systems are categorised as quite different entities. Embryologically, ureteral formation precedes and signals the formation of the kidneys through a complex interaction between the ureteric bud, Wolffian duct structures, metanephric blastema, and nephrogenic blastema. Imbalanced crosstalk among these actors leads to ectopic ureteral positioning and potential associated nonfunctioning or a dysplastic ipsilateral kidney, as well as other genitourinary abnormalities [19,20]. Finally, ectopic ureters result from the earlier development of the ureteric bud, which opens outside the bladder trigone and grows into an immature nephrogenic blastema [13].

Ectopic ureters may be identified as an SSEU, but most cases arise in patients with duplex kidneys. When ectopy occurs in patients with single ureters, such as in the case of our patient, one may speculate this diagnosis is less accessible and less frequent and therefore could more easily be overlooked. Diagnosis is often delayed because the ectopic ureter is difficult to be identified by imaging, the corresponding kidney is frequently small, dysplastic, poorly functioning, and non-visualised, and the infant may be misdiagnosed with a contralateral normal “solitary kidney”. SSEU is more commonly associated with systemic malformations [21]. Several authors have noted that the more remote the ectopic orifice lies from its normal opening the more severe are the associated renal anomalies [22,23].

In the same respect, it appears that ectopic ureters should be suspected in female foetuses with OHVIRA syndrome (obstructed hemivagina with ipsilateral renal agenesis) patients, despite the initial appearance of renal agenesis [24,25,26], and male foetuses with Zinner syndrome [27,28].

In females with an ectopic ureter, the most common sites of ureteral insertion include the bladder neck and upper urethra (33%), the vaginal vestibule (33%), the vagina (25%), the uterus (<5%), and the cervix (<5%) [29].

Several theories and rules have emerged to explain the positioning of ectopic ureters. The Mackie–Stephen hypothesis postulates that a lateral origin of the ureteral bud in the embryonic Wolffian duct will lead to a caudal orifice, whereas a medially displaced origin translates to a more upward insertion [30]. The *trigone precursor* theory also tries to disentangle the origin of ectopic ureters. When the trigone precursor—the medial portion of the Wolffian duct which joins the mesonephric duct’s remnant near the insertion of the ureteric bud—is longer, it will lead to a caudal ectopy, whereas a shorter trigone precursor will trigger a medial origin of the ureteral bud and an increased risk of ureteral reflux in affected patients [20].

In patients with duplex drainage systems, the ectopic ureter typically follows the Weigert–Meyer principle, where the ureter draining the upper moiety is the one with ectopic insertion, often leading to urinary incontinence, while the lower moiety ureter inserts normally into the bladder [31].

### 3.2. Genetic Considerations

Genetic testing during the antenatal period, coinciding with the ultrasound detection of CAKUT, is essential to rule out hereditary disorders or genetic syndromes, and to guide management and counselling. In our case, both prenatal and postnatal genetic tests yielded negative results for the genes associated with polycystic kidney disease or structural anomalies of the genitourinary system.

The most comprehensive genetic examination tool—WES—is a diagnostic means accessible both for the prenatal and postnatal investigation to identify the molecular aetiology in individuals with CAKUT, including single nucleotide variants and copy number variants (SNVs, CNVs). This can be provided as a single analysis for the offspring or as Trio WES to include parental testing.

In patients with ureteral ectopy and associated kidney abnormalities, defects in the RET (rearranged during transfection), FGFR2 (fibroblast growth factor receptor 2), and GATA3 genes have been reported, and suspected patients should be investigated for their presence [32,33,34].

### 3.3. Diagnostic Considerations

The detailed ultrasonographic foetal morphology is an excellent tool for the detection of CAKUT, but this rare ectopic ureteral opening abnormality is frequently missed, especially in cases of single ureter systems. When suspicion arises, the adjunct of prenatal MRI helps establish a diagnosis and plan for the integrative postpartum follow-up.

Few reports of the antenatal detection of ectopic ureters can be found in the literature, usually when indirectly following examination of a cystic pelvic mass, in the context of a mullerian abnormality, or a multicystic dysplastic kidney. The following entities must be considered in the differential diagnosis following the ultrasound detection of a cystic pelvic mass: ovarian cyst and hydrocolpos/hydrometrocolpos in female foetuses, the presence of a cloaca, the megacystis–microcolon–intestinal hypoperistalsis syndrome, lower urinary tract obstructions (post-urethral valves, urethral stenosis, urethral atresia), enteric duplication cysts, mesenteric cysts and lymphangiomas, anterior sacral meningoceles, sacrococcygeal teratomas, and meconium peritonitis.

Krishnan et al. presented the antenatal ultrasound detection of the ectopic ureteral drainage of an ectopic dysplastic kidney detected as a pelvic mass in a foetus with obstructed vaginal duplication [35]. Another Müllerian abnormality associated with an ectopic ureter with prenatal detection was reported by Coroado et al. (2019), further confirming the more accessible prenatal ultrasound pick-up in this context [36].

Cassart et al. reported on a series of 16 cases of multicystic dysplastic kidney associated with ureteral ectopy, of which 4 were detected prenatally, 2 by ultrasound, and 2 by MRI [37]. All structures appeared as cystic masses in the pelvis.

Ureterocoele can be visualised as a cystic pelvic mass, both by ultrasound and MRI, thus making diagnosis more accessible. Several cases of prenatal ultrasound pick-up have been reported in the literature, dating back as far as 1983 [38].

In 2003, Caire et al. described a first case of ectopic ureterocoele diagnosed by antenatal MRI in a foetus with renal duplication [39]. Chen et al. reported the MRI diagnosis of an ectopic ureter with ureterocoele at 36 weeks of gestation. Imaging was performed due to the inability of ultrasound to establish the origin/diagnosis of a foetal cystic pelvic mass. It turned out to be a duplex left kidney with a giant blind ectopic ureter arising from the upper moiety which was confirmed postpartum [40].

In cases which do no benefit from prenatal diagnosis/suspicion, the nonspecific postnatal symptoms and the absence of urinary continence in the first months often delay the diagnosis of this malformation until childhood or even adulthood [7,29,41]. Postnatal detection rates of dysplastic kidney and ectopic ureter insertion vary among the different imaging studies, being around 17.5% for ultrasound, 13.6% for computer tomography, and 33.3% for MRI [42].

Although antenatal MRI improves diagnostic suspicion, and can be safely used during pregnancy (without a contrast agent) [43,44], confirmation is only possible after birth through urethrocystoscopy, urography, magnetic resonance urography, or renal scintigraphy [7,29,41].

Magnetic resonance urography (MRU) provides detailed anatomical images of the urinary tract without exposing the patient to ionizing radiation while scintigraphy assesses both the renal structure and function but provides less detailed anatomical imaging compared to MRU. However, scintigraphy is considered the gold standard for measuring split renal functions and detecting urinary tract obstructions.

The endoscopic examination provides superior diagnostic possibilities for ectopic ureter insertion, being able to identify the aberrant ureteral orifice in up to 70% of cases according to several authors [42].

### 3.4. Outcome & Management Considerations

Abnormal ureteral insertions often result in continuous urinary incontinence, as the ectopic ureter bypasses the normal sphincter mechanism, or may lead to vesicoureteral reflux, depending on the insertion site. Identifying the specific location of insertion is crucial for diagnosis and for planning the timing and type of surgical intervention. Recurrent urinary tract infections in otherwise healthy patients may be the only clue in front of such an intricate diagnosis in infant and adults.

The management of an ectopic ureter is determined by factors such as symptoms, renal function, patient age, and quality of life. Treatment options range from conservative management in asymptomatic cases to surgical intervention when significant symptoms, recurrent infections, or impaired renal function are present. The choice of surgery depends on the renal unit’s functionality, with options including ureteral reimplantation, ureteroureterostomy, heminephrectomy, or nephrectomy for nonfunctional kidneys [7,42,43,44].

From work-up to diagnosis, including disease monitoring and interventions, ectopic ureters represent a multidisciplinary puzzle. It involves the cooperation of the obstetrician, foetal radiologist, neonatologist, paediatrician, paediatric nephrologist, and urologist. Timely diagnosis is mandatory to counsel parents regarding foetal/child outcome, and to plan the therapeutic approach. The case-tailored management can be arranged according to specifics, and early diagnostic suspicion represents a prerequisite for an accurate strategy.

## 4. Conclusions

Diagnosing an ectopic ureter opening prenatally poses a great challenge to the obstetrician, and it depends on the presence of ultrasonographic and MRI signs, which vary according to foetal sex and the ectopic implantation site. Caution must be undertaken when performing a second and third trimester foetal morphology scan to detect early signs and to perform prenatal MRI if needed for confirmation of diagnosis. Early detection allows for a patient-tailored approach in a multidisciplinary team from the antepartum and following into the postpartum period. An increased level of suspicion must be maintained postpartum in children with unspecific symptoms, such as frequent UTIs, urinary incontinence, and dribbling, and no other etiologic explanation.

## Figures and Tables

**Figure 1 diagnostics-15-01788-f001:**
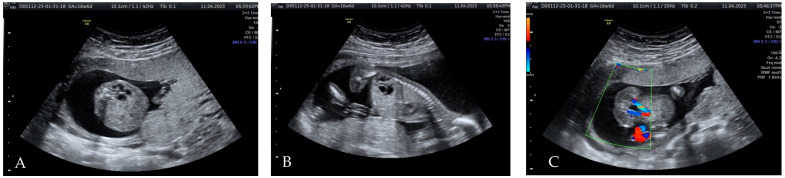
Ultrasound findings at the 17 WG scan: (**A**,**B**)—dysplastic left kidney; (**C**)—foetal bladder.

**Figure 2 diagnostics-15-01788-f002:**
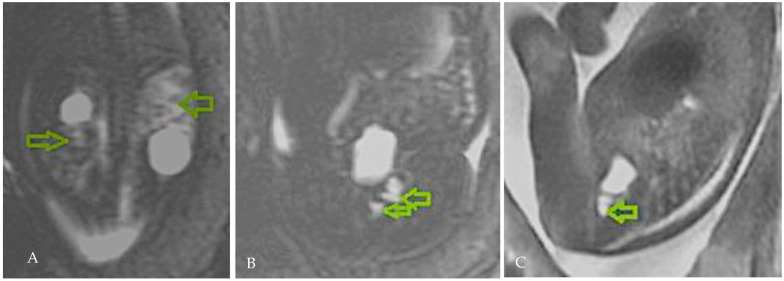
(**A**) T2-weighted coronal sections: right kidney with cortical cyst and normal parenchyma (lower arrow), left kidney with a macroscopic cyst and parenchyma with cystic dysplasia (upper arrow). (**B**) Subvesical fluid structures—ectopic ureter with vaginal/urethral opening (arrows). (**C**) Sagittal T2 sequence with subvesical fluid structure vagina/urethra with fluid content (arrow).

**Figure 3 diagnostics-15-01788-f003:**
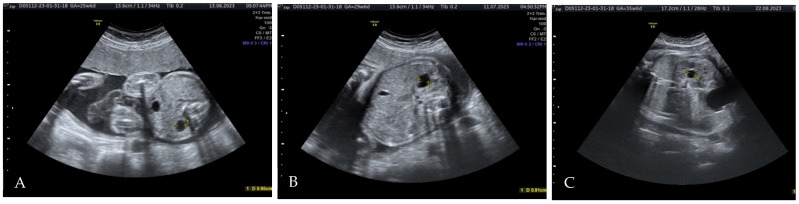
Right renal cyst at 26 WG, 30 WG and 36 WG ((**A**), (**B**), and (**C**), respectively.). Callipers are placed on each image to illustrate renal cyst dimensions.

**Figure 4 diagnostics-15-01788-f004:**
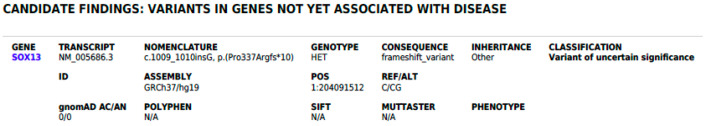
WES analysis showing only a VUS mutation in the SOX13 gene.

**Figure 5 diagnostics-15-01788-f005:**
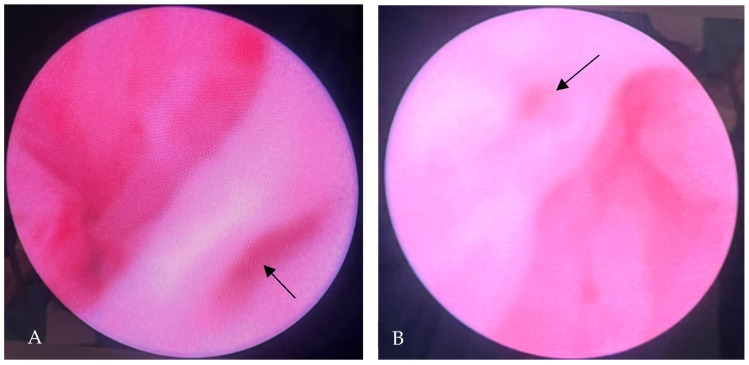
Urethrocystoscopic imaging: (**A**) the right ureteral opening at the level of the trigone (arrow); (**B**) absence of the left ureteral opening (arrow).

**Figure 6 diagnostics-15-01788-f006:**
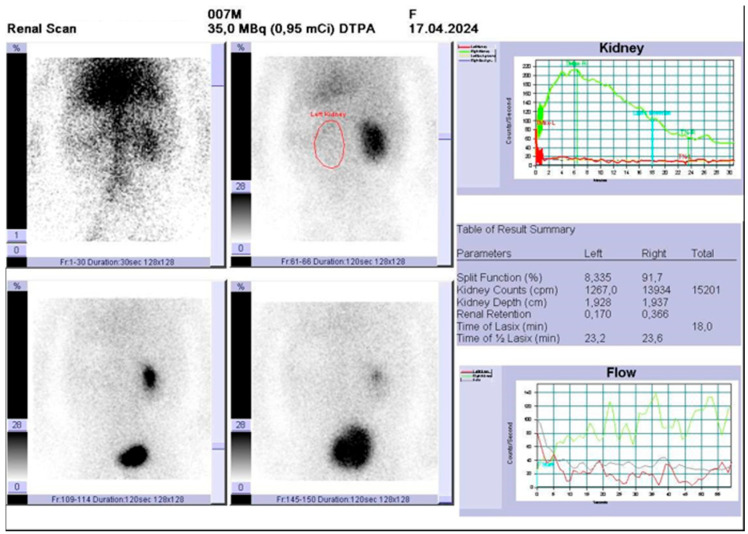
Renal scintigraphy showing absent function in the left kidney (red circle).

## Data Availability

No new data were created or analyzed in this study. Data sharing is not applicable to this article.

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
