# Peer review of "Prenatal Diagnosis of Vaginal Ectopic Ureter Insertion—Case Outcome and Literature Overview"

_diagnostics, 2025, doi:10.3390/diagnostics15141788_

Round 1
Reviewer 1 Report
Comments and Suggestions for Authors
This case report presents a rare instance of a prenatally diagnosed single-system ectopic ureter (SSEU) with vaginal insertion, confirmed postnatally and managed surgically. The manuscript is of clinical relevance due to the diagnostic complexity of SSEU and the utility of prenatal MRI in early suspicion. However, several areas require minor revision for clarity, precision, and completeness, especially in terms of grammar, sentence structure, and literature context.
- Line 11 and Line 16 should be rephrased to provide clearer and more precise explanations.
- There are several typographical and grammatical errors throughout the manuscript. A thorough proofreading is strongly recommended. For instance, in Line 13, "ecotpic" should be corrected to "ectopic."
- The conclusion of the abstract should include a statement emphasizing the significance of early diagnosis and the role of prenatal imaging in prognosis and clinical decision-making.
- The introduction contains several grammatical and typographical errors that should be corrected. First-person pronouns such as "we" and "our" should be avoided to maintain an objective tone.
- Line 32 should be rephrased to improve sentence clarity and flow.
- Consistency in terminology is essential. The manuscript alternates between "WG" and "weeks gestation"; a uniform term should be used throughout.
- Minor typographical errors need correction, such as: "reson" → "reason", "neprectomy" → "nephrectomy", "schedueld" → "scheduled"
- Line 157 is incomplete; the newborn's birth weight and Apgar score should be provided.
- A brief explanation of the whole exome sequencing (WES) methodology is missing. Since a variant of uncertain significance (VUS) was identified in the SOX13 gene, its potential clinical relevance should be discussed more thoroughly.
- Clarify whether genetic counseling or further follow-up was recommended for the parents in light of the WES findings.
- The rationale for performing the nephrectomy at 8 months should be explained to enhance the reader’s understanding of the clinical decision-making process.
- The discussion would benefit from improved structure and coherence for better continuity.
- Typographical errors such as "unill" (should be "until") and "dignosis" (should be "diagnosis") must be corrected.
- The discussion should include a broader comparison with existing literature, particularly regarding how often vaginal SSEU is diagnosed prenatally.
- Further elaboration on the diagnostic value and comparative utility of various imaging modalities (ultrasound, MRI, MRU) is needed.
- A brief overview of differential diagnoses considered during the prenatal suspicion of a cystic pelvic mass should be added.
- The misspelling of "challneging" should be corrected to "challenging."
- The conclusion should clearly reinforce the importance of accurate prenatal imaging and diagnosis for early management and better prognosis in rare congenital anomalies like SSEU.
- A patient-centred approach, highlighting timely intervention and multidisciplinary management, should be emphasized.
Author Response
Esteemed Reviewer,
Thank you for the care of your evaluation and for appreciating our manuscript.
Firstly, we apologize for all the typos and English language slip-ups, we should have read it before submitting with more attention to detail.
Otherwise, we have modified and improved our manuscript following your suggestions. All changes are highlighted in blue throughout the text.
- Line 11 and Line 16 should be rephrased to provide clearer and more precise explanations.
Line 11 - “Ectopic ureters are a rare condition, typically diagnosed in childhood and infrequently in adulthood”.
The entire abstract has been reshaped at the suggestion of another reviewer and this sentence was deleted.
Line 16 – “Early diagnosis is often challenging, as the condition may present with nonspecific symptoms, leading to underdiagnosis.”
We have rephrased as follows: ”Careful fetal scanning during the late second and third trimester might provide clues and lead to prenatal detection. However, even the postnatal diagnosis is challenging and often delayed towards adulthood since the condition may present with nonspecific symptoms, leading to underdiagnosis or misdiagnosis.”
- There are several typographical and grammatical errors throughout the manuscript. A thorough proofreading is strongly recommended. For instance, in Line 13, "ecotpic" should be corrected to "ectopic."
Everything has been addressed, thank you!
- The conclusion of the abstract should include a statement emphasizing the significance of early diagnosis and the role of prenatal imaging in prognosis and clinical decision-making.
We have appended the conclusion of the abstract with “Ectopic ureters are a pathology generating life-long morbidity and discomfort of the offspring and young adult. Awareness to this pathology must be raised among clinicians, especially regarding the potential detection by minute prenatal ultrasound examinations, followed by MRI to refine diagnosis. Postnatally, the persistence of suspicious yet unspecific symptoms, in both male and female individuals, must trigger thorough imaging/cystoscopic examination to reach diagnosis and provide correct management.”
- The introduction contains several grammatical and typographical errors that should be corrected. First-person pronouns such as "we" and "our" should be avoided to maintain an objective tone.
We apologize again for our clumsiness regarding orthographic errors. We have changed all sentences with the we/our use.
- Line 32 should be rephrased to improve sentence clarity and flow.
Line 32 - CAKUT can range from mild unilateral hydronephrosis to complex anomalies with significant impact on renal function and fetal prognosis.
We have changed this to “CAKUT represent a complex array of abnormalities with various degrees of clinical impact on the foetus, from those who require mere surveillance to those who deem medical treatment, surgery, to lethal malformations.”
- Consistency in terminology is essential. The manuscript alternates between "WG" and "weeks gestation"; a uniform term should be used throughout.
Thank you for your suggestion, we have revised all the targeted instances and, except for the first mention, changed to WG.
- Minor typographical errors need correction, such as: "reson" → "reason", "neprectomy" → "nephrectomy", "schedueld" → "scheduled"
All errors were corrected, thank you.
- Line 157 is incomplete; the newborn's birth weight and Apgar score should be provided.
We have completed the sentence, sorry for the slip-up.
“The patient delivered electively at 39WG by C-section a newborn female of 3550 g, with an Apgar score of 10
- A brief explanation of the whole exome sequencing (WES) methodology is missing. Since a variant of uncertain significance (VUS) was identified in the SOX13 gene, its potential clinical relevance should be discussed more thoroughly.
Thank you for your comment. Regarding the WES methodology technically, we did not provide a description because this is beyond the scope of the manuscript theme. Regarding the impact of abnormalities in the SOX13 gene, these are not are not incriminated for this spectrum of pathology, and the mutation identified is a VUS type (unknown significance), associations with other pathologies being difficult to affirm.
- Clarify whether genetic counselling or further follow-up was recommended for the parents in light of the WES findings.
No further counselling was recommended because this is a VUS variant.
- The rationale for performing the nephrectomy at 8 months should be explained to enhance the reader’s understanding of the clinical decision-making process.
The left kidney was non-functional, it was the decision of paediatric surgery/urology. We dare say that further explanation is beyond the scope of our presentation as a case of prenatal detection of this pathology, and beyond our exact knowledge since it is a different specialty acting at a distant time-point from the perinatal period.
- The discussion would benefit from improved structure and coherence for better continuity.
Our approach for the discussions was to talk about CAKUT as a large category, then introduce the entity of ectopic ureters. We then undertook the subject from the anatomic, to genetic implications, diagnostic considerations and then case management and outcome. We see a flow :)
We are very open to complying with your suggestion, but we need a little more guidance.
- Typographical errors such as "unil" (should be "until") and "dignosis" (should be "diagnosis") must be corrected.
We have corrected them, thank you.
- The discussion should include a broader comparison with existing literature, particularly regarding how often vaginal SSEU is diagnosed prenatally.
The entire literature regarding the prenatal detection is taken into discussion and included as bibliography since there are only these few cases. We cannot yet talk about frequency of prenatal detection.
- Further elaboration on the diagnostic value and comparative utility of various imaging modalities (ultrasound, MRI, MRU) is needed.
Our aim is to rase awareness regarding the potential prenatal detection of ectopic ureters. Given the rarity of cases detected prenatally it is hard to offer more details regarding the comparative value of ultrasound versus MRI.
When it comes to MRU, this is beyond our specialty and the scope of our review.
- A brief overview of differential diagnoses considered during the prenatal suspicion of a cystic pelvic mass should be added.
This is a very good point, thank you, we have appended the manuscript with the following text:
“The following entities must be considered in the differential diagnosis following the ultrasound detection of a cystic pelvic mass: ovarian cyst and hydrocolpos o/hydrometrocolpos in female fetuses, the presence of a cloaca, the megacystis-microcolon-intestinal hypoperistalsis syndrome, lower urinary tract obstructions (post-urethral valves, urethral stenosis, urethral atresia), enteric duplication cysts, mesenteric cysts and lymphangiomas, anterior sacral meningoceles, sacrococcygeal teratomas, meconium peritonitis.”
- The misspelling of "challneging" should be corrected to "challenging."
We have corrected the spelling error, thank you.
- The conclusion should clearly reinforce the importance of accurate prenatal imaging and diagnosis for early management and better prognosis in rare congenital anomalies like SSEU.
We have changed the entire conclusion to:
“Diagnosing an ectopic ureter opening prenatally poses a great challenge to the obstetrician and it depends on the presence of ultrasonographic and MRI signs, which vary according to fetal gender and the ectopic implantation site. Caution must be undertaken when performing second and third trimester fetal morphology scan to detect early signs and perform prenatal MRI if needed for confirmation of diagnosis. Early detection allows a patient-tailored approach in a multidisciplinary team from the antepartum and following into the postpartum period. An increased level of suspicion must be maintained postpartum in children with un-specific symptoms such as frequent UTIs, urinary incontinence and dribbling and no other etiologic explanation.”
- A patient-centred approach, highlighting timely intervention and multidisciplinary management, should be emphasized.
Thank you for the suggestion, we have introduced the following text at the end of discussions:
“From work-up, to diagnosis, disease monitoring and interventions, ectopic ureters represent a multidisciplinary puzzle. It involves the cooperation of the obstetrician, fetal radiologist, neonatologist, paediatrician, paediatric nephrologist and urologist. Timely diagnosis is mandatory to counsel parents regarding fetal/child outcome and plan the therapeutic approach. The case tailored management can be arranged according to specifics and early diagnostic suspicion represents a prerequisite for an accurate strategy.”
Reviewer 2 Report
Comments and Suggestions for Authors
The authors reported a case report of a prenatal diagnosis of Vaginal Ectopic Ureter Insertion.
This case is interesting , well documented and consistent with the literature analysis.
Here are some comments :
- improve abstract to add more details on the case presentation
- Add in the introduction the main clinical guidelines on this procedure
- Hormonal and biological samplings could be added in the case presentation
- Decrease size of arrows in figure 2
- Figure 4 : no need to put the entire report ; make number of figures and tables lower and for better clarity
- Add arrow in figure 5
- Discussion contains a literature analysis , it would be better if the authors put their different clinical and genetic aspects in a table for a better clarity
Author Response
Esteemed reviewer, thank you very much for positively evaluating our work. We have appended the manuscript in correspondence to your suggestions, on a point-by-point basis, and all changes are marked in yellow throughout the text.
- improve abstract to add more details on the case presentation
Thank you for pointing this out, we have completely reshaped the entire abstract as follows:
“Ectopic ureters are a rare urinary tract malformation, typically diagnosed in childhood and infrequently in adulthood. The prenatal detection by ultrasound and magnetic resonance imaging (MRI) of this clinical entity has scarcely been reported. Careful fetal scanning during the late second and third trimester might provide clues and lead to prenatal detection. However, even the postnatal diagnosis is challenging and often de-layed towards adulthood since the condition may present with nonspecific symptoms, leading to underdiagnosis or misdiagnosis. In female patients approximately 25% of ectopic ureters open into the vagina. Due to the high risk of recurrent urinary tract in-fections and the potential development of uretero-hydronephrosis timely diagnosis is essential, and prompt surgical correction is mandated.
We report the case of a 33-year-old GII PI patient diagnosed with cystic dysplasia of the left fetal kidney at the 16 WG (weeks of gestation) scan. The malformation was consistent at 21 WG when karyotyping by amniocentesis identified a normal female molecular karyo-type. MRI performed at 28 weeks confirmed the left renal dysplasia and raised the sus-picion of an abnormal insertion of the left ureter into the vagina. After delivery, the vaginal ureteral ectopy was confirmed at 3 weeks postpartum postnatally via cystoscopy. Postpartum whole exome sequencing identified a variant of uncertain significance (VUS) mutation in the SOX 13 gene (SRY-box transcription factor 13). Renal scintigraphy per-formed 7 months postnatally identified a hypo/afunctional left kidney which led to the indication of nephrectomy by the paediatric urologist. The surgical intervention was performed at 8 months postpartum with a favourable outcome.
Ectopic ureters are a pathology generating life-long morbidity and discomfort of the offspring and young adult. Awareness to this pathology must be raised among clinicians, especially regarding the potential detection by minute prenatal ultrasound examinations, followed by MRI to refine diagnosis. Postnatally, the persistence of suspicious yet un-specific symptoms, in both male and female individuals, must trigger thorough imaging/cystoscopic examination to reach diagnosis and provide correct management.”
- Add in the introduction the main clinical guidelines on this procedure
This is a very rare clinical entity, and the prenatal diagnosis consists of only a few cases, all of which have been cited during discussion section under the subchapter “Diagnostic considerations”. There are no available guidelines and we daresay there will not be in the near future. Our aim is to raise awareness to such a potential occurrence during fetal ultrasound such that an informed and trained eye could raise the diagnostic suspicion.
- Hormonal and biological samplings could be added in the case presentation
Since this is a urinary tract abnormality and its impact refers to the urinary tract function, there are no hormonal or biochemical markers related to this pathology to aid in the diagnosis or follow-up.
- Decrease size of arrows in figure 2
We do not have access to this figure anymore, the patient had provided it from the radiologist from another service, as such, with the arrow on it.
- Figure 4 : no need to put the entire report ; make number of figures and tables lower and for better clarity
We shrunk the figure, thank you. This was truly too much.
- Add arrow in figure 5
It was added, thank you.
- Discussion contains a literature analysis; it would be better if the authors put their different clinical and genetic aspects in a table for a better clarity.
Unfortunately, the cases reported in the literature about the prenatal diagnosis of this pathology are rare and usually consist of ectopic opening and pyelocaliceal duplication, or ureterocele, which is why we do not believe that a summary as a table form would be beneficial. Most articles focus primarily on postnatal diagnosis and therapeutic decisions.
Reviewer 3 Report
Comments and Suggestions for Authors
I had the pleasure of reviewing the paper titled: Prenatal Diagnosis of Vaginal Ectopic Ureter Insertion - case outcome and literature overview.
The authors correctly approached the issue by presenting us with their case report and literature review.
I have a few comments on the manuscript:
Line 61: MRI can offer additional diagnostic information without risk to the fetus 0 what do you mean by that?
Line 73: Pregnancy was correctly managed? Elaborate or what did you have in mind here?
Figure 2 is in rather poor quality can you provide a better resolution / quality?
Figure 5 – readers may not be related to your work, please provide “markers” within figure A/B to match description.
Line 253 – I would tone down enthusiasm about CAKUT detection in first trimester since only some are detectable at this point.
Line 312 WES or Trio WES for parents as well.
Comments on the Quality of English LanguageAlthough the work is interesting it needs to go through linguistic corrections. It may be worth having the work revised by a medical translator if it has not already been done.
Author Response
Esteemed reviewer, we truly appreciate your evaluation of our work, it is very rewarding to receive such an evaluation. Thank you for your very just comments, we have appended the manuscript as suggested and highlighted all the changes in green.
Line 61: MRI can offer additional diagnostic information without risk to the fetus 0 what do you mean by that?
We have completely removed this phrasing. It was a clumsy attempt to say that MRI is safe during pregnancy. But this is in fact an acknowledged reality, so we deleted it.
Line 73: Pregnancy was correctly managed? Elaborate or what did you have in mind here?
In our country some patients attend the routine prenatal visits and perform the required analysis and ultrasound check-ups, while others don’t. We actually use this “Incorrectly managed pregancy” as a diagnosis. This was a patient who had done everything according to recommendations.
We have changed the text to “The patient attended all the recommended prenatal visits and performed the required investigations during the first trimester. First trimester genetic screening for Down syndrome by the integrated test was performed and provided a low-risk result.”
Figure 2 is in rather poor quality can you provide a better resolution / quality?
We do not have access to this figure anymore, the patient had provided it from the radiologist from another service, as such, we have managed only minimal change.
Figure 5 – readers may not be related to your work, please provide “markers” within figure A/B to match description.
Thank you, arrows were added accordingly.
Line 253 – I would tone down enthusiasm about CAKUT detection in first trimester since only some are detectable at this point.
You are very right in your observation, maybe we did get carried away. But we did write “some anomalies”. We were thinking Prune Belly, renal agenesis, multicystic kidneys for instance. Should we still rephrase?
Line 312 WES or Trio WES for parents as well.
Thank you for pointing this out, we have added the following sentence to the manuscript in the respective paragraph: …“This can be provided as a single analysis for the offspring or as Trio WES to include parental testing.” (Line 307)